

# Optimized hybrid SVM-RF multi-biometric framework for enhanced authentication using fingerprint, iris, and face recognition

Sonal[1], Ajit Singh[2] and Chander Kant[3]

[1] Department of Computer Science & Engineering and Information Technology, Uttarakhand Technical University, Dehradun, Uttarakhand, India
[2] Department of Computer Science & Engineering, Bipin Tripathi Kumaon Institute of Technology, Almora, Uttarakhand, India
[3] Department of Computer Science & Applications, Kurukshetra University, Kurukshetra, India

## ABSTRACT

This article introduces a hybrid multi-biometric system incorporating fingerprint, face, and iris recognition to enhance individual authentication. The system addresses limitations of uni-modal approaches by combining multiple biometric modalities, exhibiting superior performance and heightened security in practical scenarios, making it more dependable and resilient for real-world applications. The integration of support vector machine (SVM) and random forest (RF) classifiers, along with optimization techniques like bacterial foraging optimization (BFO) and genetic algorithms (GA), improves efficiency and robustness. Additionally, integrating feature-level fusion and utilizing methods such as Gabor filters for feature extraction enhances overall performance of the model. The system demonstrates superior accuracy and reliability, making it suitable for real-world applications requiring secure and dependable identification solutions.

## INTRODUCTION

Biometrics are the characteristic biological measurement used to identify or authenticate the individual based on its characteristics. This approach increasingly establishes person recognition in various applications (*Boubchir & Daachi, 2021*). Although biometric recognition techniques can be very effective, they cannot guarantee an outstanding recognition rate by uni-modal biometric systems that rely on a single biometric signature (*Rasheed et al., 2023*). Sensor noise, non-universality, a lack of uniqueness and consistent representation, and sensitivity to attack are all common problems with these systems (*AlRousan & Intrigila, 2020*). Because of these practical constraints, the mistake rates connected with uni-modal biometric systems are relatively high, making them inappropriate for deploying vital security applications (*Bharadwaj, Vatsa & Singh, 2014*; *Ammour et al., 2020*). A technique known as mobile biometric service is implemented in various biometric modalities in the same system to address these issues. This study offered

Corresponding author
Sonal, sonalkharb@gmail.com

a multi-modal identification method that combined facial, iris, and fingerprint data (*Hamd & Mohammed, 2019*). Given the difficulty distinguishing people, especially in massive databases, face recognition has played a vital role in bringing out more researchers' interest in this domain (*Yang, 2024*). The recognition process can be classified based on emotion, age, and gender (*Momin & Tapamo, 2016*). As a result, this is viewed as the unchangeable and most dominant human feature that, unlike gender and age, could be easily concealed, even under camouflage (*Connie et al., 2017*). Face recognition based on emotion has gained popularity in recent years for a range of applications, including improving surveillance systems and safeguarding various types of multimedia information (*Manesh, Ghahramani & Tan, 2010*; *Lakafosis et al., 2011*), personal privacy and natural identity management (*Xie, Hu & Wu, 2019*), and controlling industrial systems (*Karau et al., 2015*). Moreover, categorization based on emotion, age, and gender can be used in various new applications, such as mobile multimedia security and determining crowd composition.

Biometric verification and identification technologies have gained more popularity in recent years, resulting in the technology's widespread use (*Gupta, Buriro & Crispo, 2018*). Most importantly, laptops with fingerprint readers and the Windows 10 "hello" feature that support biometric verification and identification are frequently seen (*Charfi et al., 2017*; *Alpar & Krejcar, 2018a*; *Nguyen, Tay & Chui, 2015*). Users who register for biometric usage are eligible for the latter functionality. The stress of logging into devices and products (like cards or keys), often with identity and privacy theft risks, is eliminated by biometric authentication (*Zhang, Cai & Zhang, 2017*). Collecting a person's biometric information to generate their biometric template is intricate and occasionally results in an illogical result (*Carroll et al., 2021*). However, the success rate of these systems might vary from up to 99 percent in the finest systems. Therefore, biometric solutions help reduce security issues, including identity theft and privacy invasion (*Alpar & Krejcar, 2018b*). When a person reaches adulthood, the characteristics of their hands are fixed for the rest of their lives. Therefore, these traits can be utilized to recognize and validate a person (*Panda et al., 2021*). Many biometric systems currently in use function based on the hand's surface and shape elevation and are used to identify a person (*Abbas et al., 2020*; *Renukalatha & Suresh, 2018*). Implementing such systems has several advantages, especially given how simple it is to obtain the necessary hardware and software (*Yang et al., 2023*). However, the biometric properties of a hand's contour are frequently insufficient to identify between people. Due to compromised identity security, there is a high false-match rate. Consequently, improved models are constantly created based on unique, concealed biometric traits that are impossible to duplicate. For instance, the findings of cutting-edge fingerprint scanning are duplicated, but its design and implementation costs are very costly (*Yin et al., 2023*). Employing two or more biometric traits within a system is a second option for enhancing verification and identification by biometric systems (*Waluś, Bernacki & Konopacki, 2017*). For example, simultaneous hand and bloodstream scanning improves identification findings and has recently become more affordable and secure (*Singh, Singh & Ross, 2019*). The fact that the false matching rate of the various biometric variables is compounded reduces the false matching rate, which makes this possible.

Uni-modal biometric systems, while effective in controlled environments, face significant limitations such as sensor noise, lack of uniqueness, and high error rates due to non-universality and inconsistent trait representation (*Al-Dabbas, Azeez & Ali, 2024b*; *Murshed et al., 2023*). These challenges make them unsuitable for high-security applications as they often fail to provide consistent and reliable performance, particularly in environments with varying conditions or when addressing large and diverse populations, necessitating the adoption of multi-modal biometric approaches to enhance accuracy, robustness, and security (*Balti et al., 2024*; *Cherrat, Alaoui & Bouzahir, 2020*). Multi-biometric approaches are more effective because they combine the strengths of different biometric traits, leading to higher performance and more reliable security. A hybrid model that combines support vector machine (SVM) and random forest (RF) classifiers is employed for such application was based on the strengths of these machine learning (ML) techniques (*Gawande, Zaveri & Kapur, 2013*). SVM is known for its ability to perform well with high-dimensional data and to find the optimal hyperplane for classification tasks (*KaviPriya & Muthukumar, 2018*). On the other hand, RF is effective for handling complex datasets and offers advantages in terms of handling noise and preventing overfitting through its ensemble learning approach (*Li, Xie & Bin, 2024*). By combining these classifiers in a hybrid framework, the study aimed to leverage the strengths of both methods to enhance classification accuracy and reduce error rates. The suggested approach addressed key challenges in biometric authentication, including accuracy, system robustness, and computational efficiency, making the approach suitable for real-world multi-biometric applications.

This study presents a novel hybrid multi-biometric system that uniquely integrates fingerprint, face, and iris recognition with advanced ML techniques to enhance biometric authentication accuracy and reliability. Unlike conventional systems, the proposed approach employs a hybrid classification model combining SVM and RF classifiers, further optimized using bacterial foraging optimization (BFO) and genetic algorithms (GA). Additionally, the use of feature-level fusion and Gabor filters for precise feature extraction enhances system performance by addressing the inherent limitations of uni-modal biometric systems, such as sensor noise and high error rates. This comprehensive methodology demonstrates a significant improvement in accuracy and robustness, providing a state-of-the-art solution for secure and reliable authentication in real-world applications.

The structure of the article is systematically organized into several critical sections for clarity and a thorough examination of the topic. The article introduces a hybrid multi-biometric system using fingerprint, face, and iris recognition, enhanced by SVM-RF model for improved performance and security in practical applications. The second section is dedicated to a comprehensive literature review, providing context and background by exploring existing research and related studies. The third section delves into the proposed methodology, offering a detailed explanation of the techniques and approaches utilized in the study. The fourth section presents the results, critically discusses the study's outcomes, and compares them with previous findings. This section is essential for evaluating the effectiveness and implications of the proposed system. Finally, the study concludes in the

fifth section, summarizing the essential findings and contributions and suggesting potential directions for future research.

## LITERATURE REVIEW

*Barra et al. (2019)* developed a mobile hand identification system using a device's camera to capture hand images in visible light, extracting 57 features through hand segmentation. This two-phase method first employs dimensional reduction to isolate robust features, then contrasts various matching techniques, achieving an equal error rate (EER) of 0.52 percent, indicating improved processing efficiency and accuracy by removing distortion-prone features. *Veluchamy & Karlmarx (2017)* designed a multi-modal biometric system by fusing finger vein and knuckle images using a fractional firefly (FFF) optimizer. They extracted features *via* a repeated line-tracking approach and optimized weight scores for feature fusion, employing a layered k-SVM classifier for recognition. This system demonstrated a 96% accuracy rate, evaluated by accuracy, false rejection, and false acceptance ratios. *Saadat & Nasri (2015)* developed a novel multi-instance human identification model with the help of various finger vein biometrics. The score level fusion is used to execute the multi-biometric system. The efficiency of various score-level fusion strategies is thoroughly investigated. The implementation results show that the highest score level fusion strategy performs better than other fusion techniques and some traditional fusion techniques for biometric models that have been created.

*Wang, Zhang & Shark (2014)* created a biometric detection system using hand vein images captured with near-infrared imaging, focusing on crucial point matching and employing geometric rectification, image enhancement, and region-of-interest extraction to handle an extensive database with over 200 classes. Based on measurements of a user's hand geometry and vascular pattern, *Park & Kim (2013)* developed a hand biometric identification system. The angles and lengths of finger valleys, profiles and lengths of the fingers, K-curvature with the hand-shaped code chain, extraction method of direction-based pattern, and lengths and angles of the hand geometry employed to obtain the hand geometry, that helps in creating a new multi-modal biometric approach. The feature points in the multi-modal biometric model are extracted from a single image. The performance of the multi-modal biometric approach to vascular pattern recognition and hand geometry is measured at the score level. The findings indicated that the developed system's equal error rate was 0.06 percent. *Kang & Park (2010)* proposed a multi-modal system integrating score-level finger geometry with vein detection, utilizing Fourier descriptors for robust finger recognition and SVM-based score-level fusion, significantly reducing error rates compared to single-method approaches. The literature survey summary is depicted in Table 1.

## RESEARCH METHODOLOGY

The study proposes a multi-biometric system integrating face, iris, and fingerprint validation to enhance the accuracy of individual authentication. The methodology is systematically divided into several key phases: dataset loading, data pre-processing, feature

**Table 1 Summary of the literature review.**

| Author(s) | Methods | Sample size | Error (%) |
|---|---|---|---|
| *Barra et al. (2019)* | FG and PP | 100 | 0.52 |
| *Veluchamy & Karlmarx (2017)* | FV and FK | 100 | 0.35 |
| *Saadat & Nasri (2015)* | MFV | 106 | 0.08 |
| *Wang, Zhang & Shark (2014)* | FV and HG | 204 | 0.02 |
| *Park & Kim (2013)* | HG and FV | 100 | 0.06 |
| *Kang & Park (2010)* | FG and FV | 102 | 0.074 |

Note:
FG, Finger geometry; PP, palm print; FV, finger vein; FK, finger knuckle; MFV, multi-finger vein; HG, hand geometry.

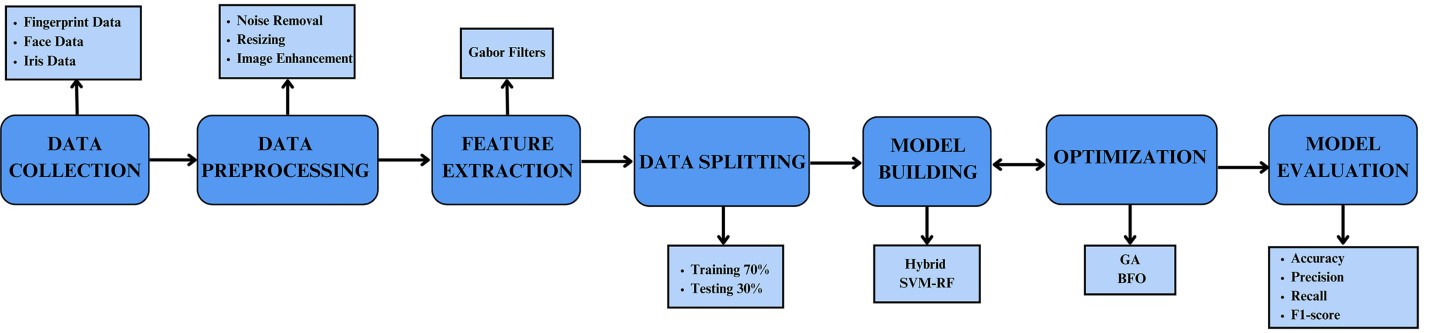

**Figure 1 Schematic illustration of the proposed model.**

extraction, classification using the proposed hybrid SVM-RF classifier, and optimization using GA and BFO. The final phase involves comparing the results with state-of-the-art techniques to evaluate performance improvements. The basic schematic of the developed methodology is illustrated in Fig. 1.

## Dataset used

In this study, three distinct datasets, including fingerprint, face, and iris data, are utilized to implement the proposed methodology. The datasets are processed uniformly to ensure comprehensive evaluation and validation, accounting for variations in biometric traits.

## System configuration

The computing infrastructure used in this study included MATLAB software (The MathWorks, Natick, MA, USA) for algorithm implementation and data analysis. The experiments were conducted on a system running a Windows 10 operating system, equipped with an Intel Core i7 processor and 16GB of RAM. This setup provided adequate computational power to handle the data preprocessing, feature extraction, and classification tasks for the multi-biometric datasets, ensuring efficient execution of the proposed hybrid SVM-RF model. The MATLAB environment facilitated integration of various ML and optimization algorithms.

## Data pre-processing

The datasets undergo a comprehensive pre-processing phase designed to standardize inputs, enhance quality, and remove noise for effective feature extraction. The pre-processing techniques includes:

### Noise removal

To reduce noise in fingerprint and iris images, median filtering with a kernel size of $3 \times 3$ is applied, which effectively mitigates salt-and-pepper noise while preserving critical edge details.

### Image resizing

All images are uniformly resized to standardized dimensions of $128 \times 128$ pixels using bilinear interpolation, ensuring compatibility with the feature extraction process and reducing computational complexity.

### Contrast enhancement

For contrast enhancement, adaptive histogram equalization (CLAHE) with a clip limit of 2.0 is employed on face and iris images, which improves visibility of subtle features while preventing over-enhancement in low-contrast areas (*Huang, 2022*). These carefully calibrated steps ensure that the input data is optimized, noise-free, and feature-rich, providing a robust foundation for the subsequent phases of the methodology.

## Feature extraction using the Gabor filter method

Gabor filters (GF) are employed to extract features, leveraging their capability to analyze spatial and frequency domain information (*Shroff & Maheta, 2015*). The convolution process with Gabor filter banks is used to identify textural patterns unique to each biometric trait. The extracted Gabor features are multi-dimensional and expressive, reducing redundancies through dimensionality reduction techniques (*Muthukumar & Kavipriya, 2017*). The mathematical formulation of Gabor filters ensures robust extraction of discriminative features, suitable for handling diverse biometric traits like fingerprint ridges, iris textures, and facial structures (*Patro et al., 2020*). The convolution process of an input picture with a GF bank primarily estimates the GF as follows:

$$G_{u,v}(p,q) \;=\; I(p,q) * \psi(p,q). \tag{1}$$

These Gabor characteristics are further discernible based on a filtering action of the Gabor with size "u" and orientation (*Garg, Vig & Gupta, 2016*; *Purohit & Ajmera, 2021*). This convolution is frantically carried out for the actual and imagined parts. The initial definition of the Gabor feature representation is:

$$Re(O(p,q))_{m,n} \;=\; I(p,q) * Re(\psi(p, q, \lambda_m, \theta_n)) \tag{2}$$

$$Im(O(p,q))_{m.n} \;=\; I(p,q) * Im(\psi(x, y, \lambda_m, \theta_n)). \tag{3}$$

As a result, calculating amplitude incorporates both imaginary and genuine parts,

$$O(p, q)_{m, n} = \left( \left( Re(O(p, q))_{m, n} \right)^2 + \left( Im(O(p, q))_{m, n} \right)^2 \right)^{\frac{1}{2}} \tag{4}$$

when utilizing GF, the multi-dimensional and highly expressive Gabor features are crucial in dimensionality reduction. These Gabor features, once filtered, form a well-defined feature vector that encapsulates the essential characteristics of the input data. This feature vector is employed to identify the class of the test data for the classification task. This process is facilitated by a supervised learning algorithm trained on labeled data to accurately classify new, unseen data based on the extracted Gabor features. The use of Gabor characteristics not only enhances the efficiency of the feature extraction process but also significantly improves the accuracy of the classification task.

## Classification using SVM and RF

The choice of SVM and RF classifiers for this study was made based on their complementary strengths in handling different aspects of classification tasks. These algorithms were selected to achieve robust and accurate classification in the multi-biometric system, considering the nature of the data and the need for efficient processing. Classifiers are central to ML tasks, where their role is to categorize data based on previously learned patterns (*Ulhe et al., 2024*). In this study, classification is performed by first organizing the extracted features into feature vectors, which serve as input for the classifiers. The process involves dividing the dataset into training and testing sets to evaluate the performance of the classifiers. The selection of SVM and RF was driven by their distinct characteristics, which together enhance the system's ability to identify individuals accurately (*Prakash, Krishnaveni & Dhanalakshmi, 2020*). SVM is employed in this study due to its effectiveness in finding the optimal decision boundary, or hyperplane, that separates different classes in a high-dimensional feature space (*Kulkarni et al., 2024*). It is particularly suitable for datasets with many features, where it aims to maximize the margin between data points of different classes. The objective function of SVM can be expressed as (*Santoso, Safitri & Samidi, 2024*):

$$min \frac{1}{2} ||\omega||^2 + C \sum_{i=1}^{N} \xi_i \tag{5}$$

subject to:

$$y_i(\omega \cdot x_i + b) \geq 1 - \xi_i, \ \xi_i \geq 0, \ \forall_i \tag{6}$$

where, weight vector is denoted by $\omega$, b represents the bias term, $\xi_i$ are slack variables that allow for some misclassification, C is the regularization parameter that controls the trade-off between maximizing the margin and minimizing classification errors, $y_i$ and $x_i$ represents the class labels and input feature vectors, respectively. In this study, a linear kernel is used, which computes the decision boundary directly based on the feature space, making it computationally efficient for classification tasks.

RF was selected to complement SVM, particularly for its ability to handle larger datasets and imbalanced data distribution effectively (*Reddy et al., 2024*). RF is an ensemble

learning method that builds multiple decision trees during training and merges their outputs for a final prediction. The method constructs each decision tree using a randomly selected subset of features, which reduces overfitting and enhances generalization. The classification process in RF can be described as follows (*Shi et al., 2024*):

Each decision tree is trained using a different subset of the data. For each node, the splitting criterion is chosen to maximize the separation between classes, commonly using metrics like Gini impurity:

$$G = 1 - \sum_{i=1}^{N} p_i^2 \qquad (7)$$

where $p_i$ is the probability of class $i$.

For classification tasks, the final prediction is determined by the majority vote from all decision trees,

$$\hat{y} = mode(y_1, y_2, \dots \dots , y_T) \qquad (8)$$

where $\hat{y}$ is the predicted output and T is the total number of trees.

The hybrid algorithm combines the strengths of SVM and RF, leveraging SVM's capacity for precise decision boundaries and RF's ability to handle large, complex datasets. The approach involves initially classifying the data using SVM to achieve a high level of accuracy, followed by using RF to further refine the predictions, especially in cases of overlapping classes or noisy data. The integration of these classifiers was chosen to improve the overall performance, as SVM efficiently handles linear separability while RF's ensemble learning addresses non-linear relationships and reduces overfitting. This combination enhances the identification process, resulting in superior classification outcomes compared to using either technique alone.

## Optimization using GA and BFO

The selected features are optimized with the help of two different algorithms, the GA and the BFO. This phase is one of the methodology's most critical phases because it helps optimize the results compared to the previous methods. Artificial intelligence uses GA as a search tool, which uses the natural selection process. With the help of natural evolution, various beneficial solutions to optimization problems are developed (*Gururaj et al., 2024*). In this study, GA selects the best feature sets by considering the mutual information between the features and the output. Population ($N = 72$) is first created from the retrieved feature set. Using the mutual information-based fitness function, GA determines each feature's fitness and transforms the results into a more usable set of benefits for ML models. Based on fitness levels, it chooses the members designated as parents. In order to create a child for the following generation, crossover and mutation strategies are used. This procedure is repeated till the target satisfies the requirements. The acquired optimal feature set is iteratively tested with the classifier until a low error rate and high accuracy are achieved, with BFO and GA employed in tandem (*Shanmugasundaram, Mohamed & Ruhaiyem, 2017*).

The BFO algorithm was employed to discover the local best value ($P_{best}$). However, the BFO algorithm exhibited slow convergence, primarily due to the fixed step size during the tumbling stage (*Al-Dabbas, Azeez & Ali, 2024a*). Despite this limitation, BFO demonstrated a solid capability to avoid local optima. Consequently, BFO was utilized to identify the local best value ($P_{best}$), while GA was used to determine the global best search ($G_{best}$). The combination of these methods effectively mitigated the issue of sluggish convergence, as illustrated by the following equations.

$$\theta_i(j + 1, k, l) = \theta_i(j, k, l) + C(i) * \theta_j \qquad (9)$$
$$P_{best} = f(\theta_i(j + 1, k, l)) \qquad (10)$$

where $\theta_i$ is the new position of the i[th] position, $\theta_j$ is the previous position, $C(i)$ is the step size, and $P_{best}$ is the local best fitness value. This combination ensures both global exploration and local exploitation, resulting in optimized feature subsets that improve classification accuracy. Experimental results confirm the efficacy of this optimization strategy, reducing the error rate and improving the system's precision and recall metrics.

## Performance metrics and evaluation

The assessment metrics used in this study were carefully selected to evaluate the performance of the multi-biometric recognition system accurately and comprehensively. These metrics include accuracy, precision, recall, F1-score, and specificity. Each metric serves a distinct purpose in assessing various aspects of the system's effectiveness in classification tasks, ensuring a balanced evaluation.

TP: true positive, TN: true negative, FP: false positive and FN: false negatives.

1. *Accuracy*: This metric measures the proportion of correctly classified instances out of the total number of samples. However, in cases where there is class imbalance, accuracy alone may not fully capture the system's effectiveness, as it can be biased towards the majority class. Therefore, while accuracy is useful, other metrics are also considered to give a more complete evaluation.

$$Accuracy = \frac{TN + TP}{TN + FP + TN + FN}. \qquad (11)$$

2. *Precision*: Precision assesses the accuracy of the positive predictions made by the model, indicating how many of the samples labelled as positive are truly positive. This metric is particularly important in scenarios where false positives need to be minimized, such as in biometric security systems where incorrectly granting access could lead to security breaches.

$$Precision = \frac{TP}{TP + FP}. \qquad (12)$$

3. *Recall*: Recall, or sensitivity, measures the model's ability to identify all relevant instances within the dataset. It indicates the proportion of actual positives that are correctly classified by the system. High recall is crucial in cases where failing to identify a positive

instance (FN) could have significant consequences, such as in authentication where genuine users must not be denied access.

$$Recall = \frac{TP}{TP + FN}.$$ (13)

4. *F1-score*: The F1-score is the harmonic mean of precision and recall, providing a single metric that balances both concerns. The F1-score helps to ensure that the system performs well across different conditions, rather than optimizing for one metric at the expense of the other.

$$F - Measure = 2 * \frac{Precision * Recall}{Precision + Recall}.$$ (14)

5. *Specificity*: This metric measures the proportion of TN that are correctly identified, providing an indication of the system's ability to recognize negative instances accurately. Specificity is important in biometric systems where it is necessary to reduce the number of false alarms (FP), thus increasing the reliability of the system.

$$Specificity = \frac{TN}{TN + FP}.$$ (15)

The combination of these metrics ensures a comprehensive evaluation of the model's performance across different dimensions. While accuracy gives an overall picture, precision and recall provide insights into the handling of positive cases, and the F1-score balances precision and recall. Specificity adds another layer by evaluating the correct identification of negative cases. This multi-faceted approach allows for a thorough and balanced assessment of the multi-biometric recognition system's effectiveness.

# RESULTS AND DISCUSSIONS

The comprehensive implementation of the suggested methodology was conducted using MATLAB software. Initially, the dataset was divided into two parts: 70 percent was designated for training, while the remaining 30 percent was reserved for testing. This split ensures that the model has substantial data for learning and sufficient data for evaluation. The detailed step-by-step process and corresponding results of the developed methodology are outlined below:

## Performance metrics for various datasets

The performance metrics of individual biometric datasets (face, finger, and iris) as well as a combined multibiometric dataset is studied as it is crucial for understanding the strengths and limitations of different models in a comprehensive manner. By examining precision, recall, accuracy, and F1-score across these datasets, the model's effectiveness and robustness in various biometric recognition tasks can be determined. This analysis not only helps identify the most suitable model for specific biometric traits but also underscores the potential improvements achieved through the integration of multiple biometric modalities.

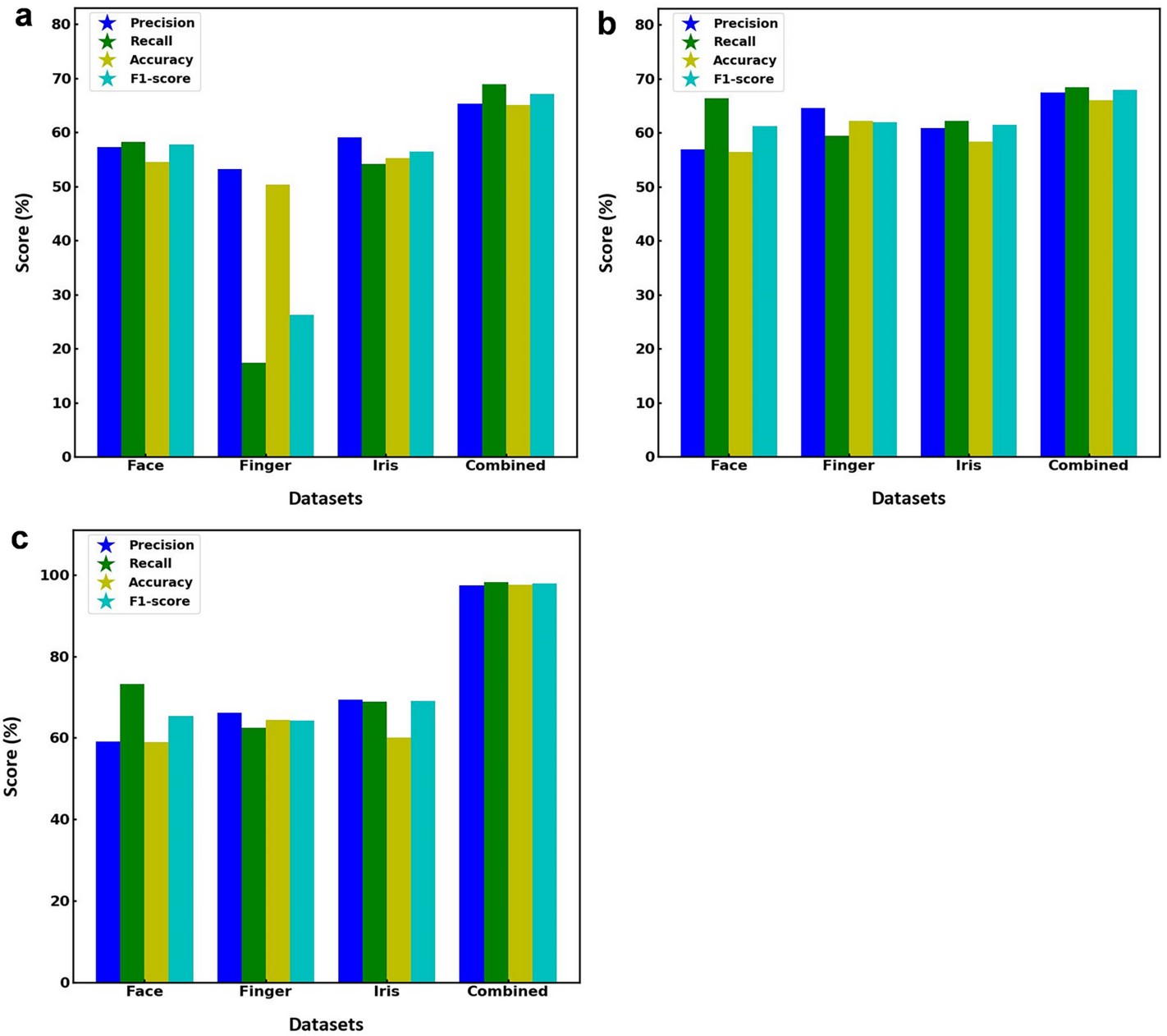

**Figure 2 Performance metrics of (A) SVM, (B) RF, and (C) hybrid SVM-RF model for various datasets.**

Figure 2 illustrates the performance metrics of three different models including SVM, RF and the proposed hybrid SVM-RF, when applied to separate face, finger, iris, and combined multibiometric datasets. Figure 2A depicts the SVM model performance for various datasets, suggesting the performance is relatively consistent around 50–60% for all face datasets. However, there is a noticeable drop in performance, particularly in accuracy, which falls below 40%, when model is applied to the finger biometric dataset. The iris

dataset performs better, with scores around 60% whereas the combined dataset yields the highest scores across all metrics, with values approaching 70%. Figure 2B displays the RF model performance showing the scores are fairly even, with all metrics hovering around 60–70% for face dataset. The finger dataset shows slightly better performance compared to the SVM model, with scores above 50%. The iris dataset maintains high scores, like the face dataset. The combined dataset once again produces the best results, with all metrics scoring around 70%. Figure 2C presents the proposed hybrid SVM-RF model performance, estimating high scores around 70% for the face dataset. The finger dataset shows an improvement over the previous models, with all metrics around 60%. The iris dataset maintains strong performance, with scores above 70%. The combined dataset significantly outperforms the individual datasets, with all metrics scoring close to 100%, indicating superior accuracy, precision, recall, and F1-score. This indicates that the hybrid SVM-RF model achieves superior performance when integrating multiple biometric modalities, thus validating the effectiveness of the hybrid approach in a multibiometric system. The model assessment parameters for SVM, RF and hybrid SVM-RF models towards various datasets is illustrated in Table 2.

The results underscore the importance of using combined multibiometric datasets for enhanced performance in biometric recognition systems. The proposed hybrid SVM-RF model consistently outperforms the individual SVM and RF models, particularly on the combined dataset, showcasing its effectiveness and potential for real-world biometric applications. This study demonstrates that leveraging a hybrid approach and integrating multi-biometric modalities can significantly enhance the performance accuracy of biometric systems.

## Performance metrics of the suggested hybrid SVM-RF model

The hybrid SVM-RF model has demonstrated superior performance compared to the individual SVM and RF models, this hybrid model is now employed on a multibiometric dataset for further analysis. The rationale behind using a multibiometric dataset lies in its potential to provide more accurate and robust results. By integrating multiple biometric traits such as face, finger, and iris, the multibiometric approach capitalizes on the strengths of each individual modality, thereby reducing the overall error rate and enhancing the system's reliability. This comprehensive evaluation allows for a more nuanced understanding of the hybrid model's capabilities, ensuring that the resulting biometric system is both precise and resilient.

As shown in Fig. 3, the training phase of the model exhibits a clear trend: as the number of epochs increases, the training accuracy improves while the error loss decreases. Figure 3A illustrates the accuracy of the models over 100 epochs. The hybrid SVM-RF model shows a consistent improvement in accuracy, starting at approximately 60% and reaching 97.56% by the 100th epoch. This suggests the effective learning and adaptation capabilities of the model. The RF and SVM model exhibit the lowest initial accuracy which has increased roughly to 54.52% and 66% by the 100th epoch, respectively. Figure 3B depicts the loss values obtained for the models across 100 epochs. The hybrid SVM-RF model shows a significant decrease in loss, starting at around 0.6 and reducing to nearly

**Table 2 Performance parameters for different models across various datasets.**

| Model | Performance metrics (%) | Face dataset | Fingerprint dataset | Iris dataset | Multi-biometric dataset |
|---|---|---|---|---|---|
| SVM | Accuracy | 54.52 | 50.26 | 55.18 | 65.03 |
| | Precision | 57.20 | 53.12 | 58.99 | 65.31 |
| | Recall | 58.25 | 17.41 | 54.11 | 68.90 |
| | F1-score | 57.72 | 26.23 | 56.44 | 67.06 |
| RF | Accuracy | 56.44 | 62.21 | 58.31 | 65.99 |
| | Precision | 56.87 | 64.59 | 60.88 | 67.45 |
| | Recall | 66.35 | 59.39 | 62.14 | 68.42 |
| | F1-score | 61.24 | 61.88 | 61.50 | 67.93 |
| Hybrid SVM-RF | Accuracy | 58.92 | 64.41 | 60.12 | 97.56 |
| | Precision | 59.10 | 66.12 | 69.34 | 97.37 |
| | Recall | 73.27 | 62.43 | 68.8 | 98.12 |
| | F1-score | 65.43 | 64.29 | 69.06 | 97.92 |

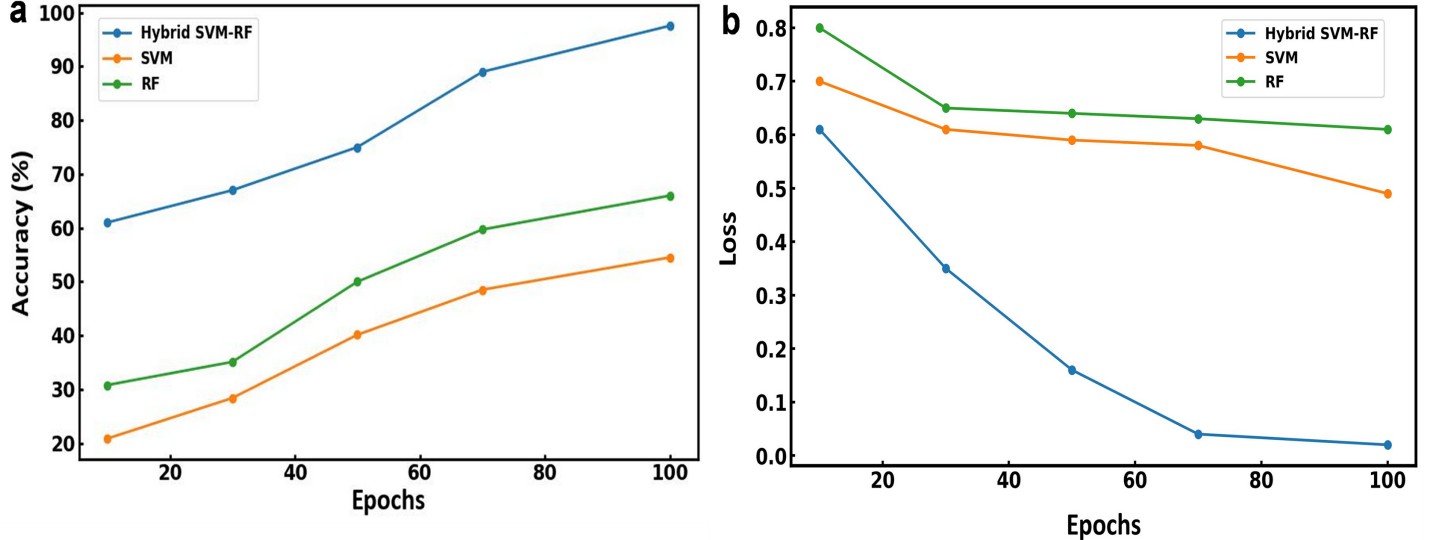

**Figure 3** (A) Accuracy and (B) loss obtained using various model for multibiometric dataset.

0.02, indicating a substantial improvement in the model's prediction accuracy. The RF model starts with the highest loss at about 0.8, which gradually decreases to around 0.64, while the SVM model begins with a loss of approximately 0.7, decreasing to about 0.49 by the 100th epoch. The results demonstrate that the hybrid SVM-RF model beats the existing models in terms of both accuracy and loss reduction when applied to combined multibiometric datasets.

The performance parameters such as precision, F1 score, and recall obtained for the models is presented in Fig. 4. Figure 4A illustrates the precision of each model over 100 epochs. Among all the models, the hybrid SVM-RF model shows a consistent

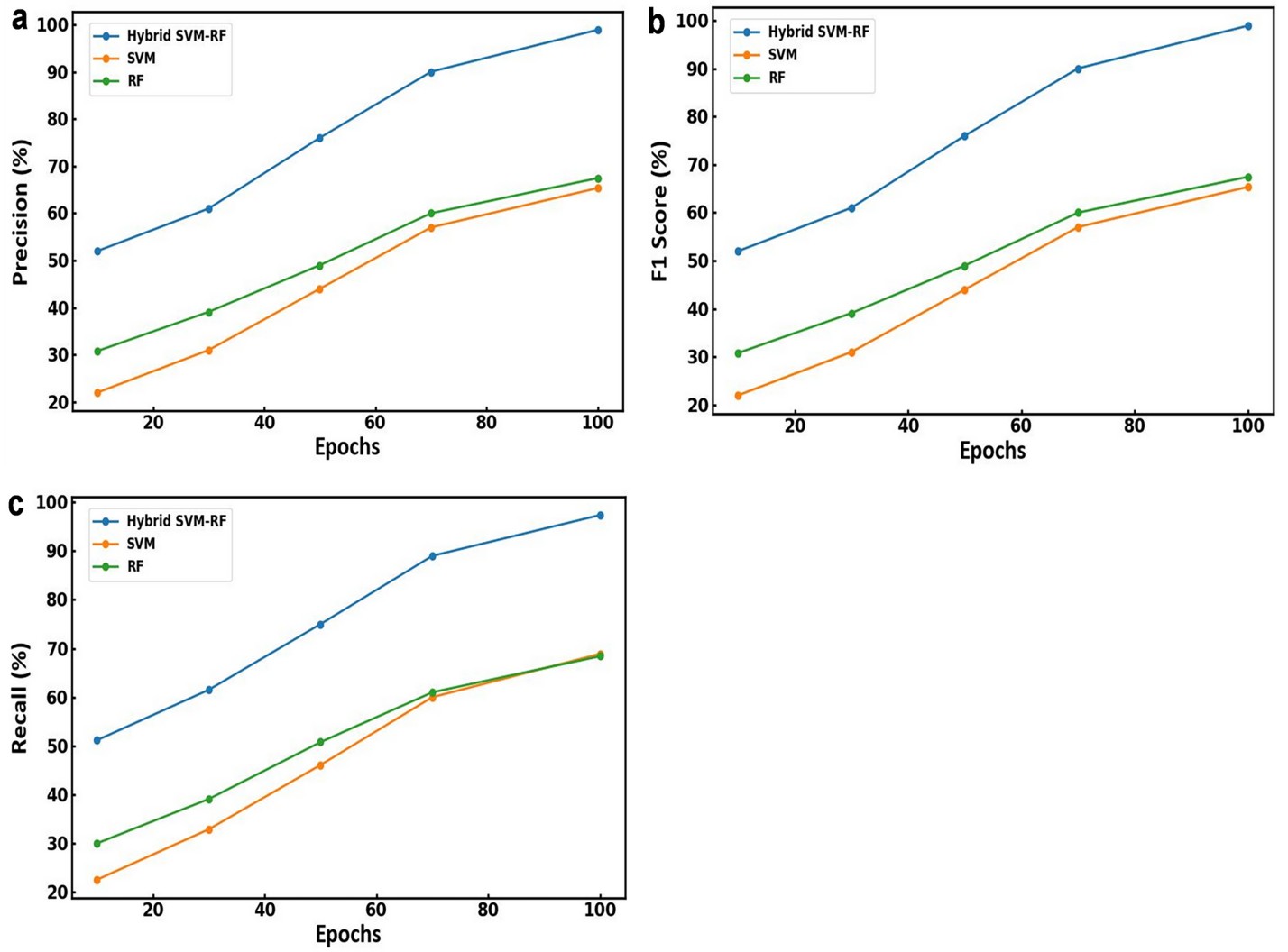

**Figure 4** (A) Precision, (B) F1 score and (C) recall obtained using various model for multibiometric dataset.

improvement in precision and evaluated around 98.89% by the 100th epoch. However, the RF and SVM model shows a steady increase in precision up to 67.45% and 65.35%, respectively by the 100th epoch. Figure 4B depicts the F1 score for each model and showed a gradual increase till 67.45% and 65.35% for RF and SVM model, respectively. Although, hybrid SVM-RF model shows a significant improvement with the F1 score increasing from around 50% to 98.12% by the 100th epoch. Similarly, the recall values for the models over 100 epochs is presented in Fig. 4C suggesting the hybrid SVM-RF model exhibits a substantial increase in recall up to 97.37% till the training period ending as compared to RF and SVM models. Therefore, the proposed hybrid SVM-RF model came out to be superior compared to both the SVM and RF models in terms of precision, F1 score, and recall when applied to combined multibiometric datasets. This highlights the robustness and

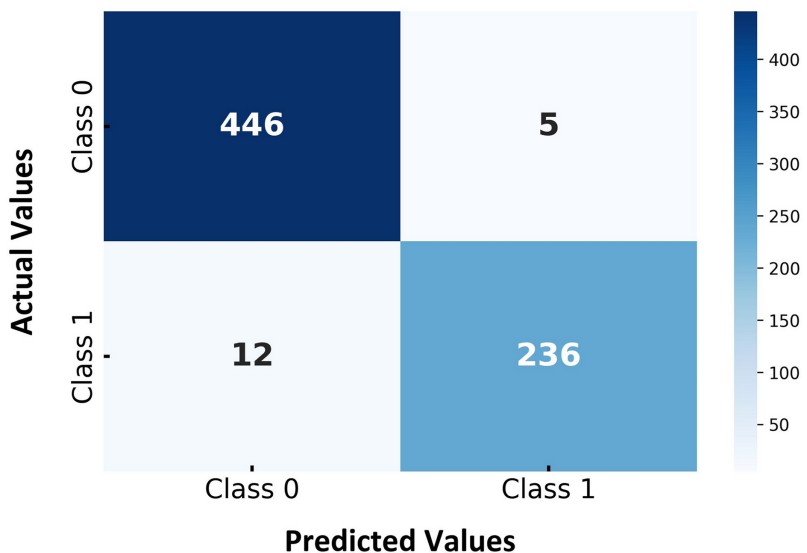

**Figure 5** Confusion matrix for hybrid SVM-RF multi-biometric system performance.

effectiveness of the hybrid approach in achieving higher performance metrics, thus validating its suitability for multibiometric recognition systems.

Figure 5 presents the confusion matrix for the proposed hybrid SVM-RF multi-biometric system, highlighting the classification performance across different classes. The matrix is divided into two target classes, with the actual class labels on the vertical axis and the predicted class labels on the horizontal axis. As shown in Fig. 5, the model correctly identified 446 instances of Class 0 whereas 5 instances of Class 0 were incorrectly classified as Class 1, representing false positives. Furthermore, the model accurately predicted 236 instances of Class 1 while 12 instances of Class 1 were misclassified as Class 0, indicating false negatives. The confusion matrix demonstrates that the hybrid SVM-RF model exhibits strong classification capabilities, achieving high accuracy rates and maintaining low misclassification rates across all classes. The relatively low numbers of false positives and false negatives further underscore the model's effectiveness in distinguishing between the two classes, thereby validating its robustness and reliability for classification tasks in multibiometric systems. The high accuracy and low error rates indicate the model's potential for practical applications where reliable biometric authentication is crucial.

Table 3 presents a detailed set of parameters obtained during implementing the proposed hybrid SVM-RF multi-biometric system. These parameters are essential for analyzing the characteristics and performance of biometric data. The measured values include a contrast of 0.3059, indicating the difference in luminance that makes objects distinguishable, and a correlation of 0.1421, reflecting the linear relationship between pixels. The energy value is 0.78623, representing the uniformity in the image's texture. At the same time, homogeneity is measured at 0.93793, showing the closeness of the distribution of elements to the diagonal in the gray level co-occurrence matrix (GLCM). The mean intensity of the pixel values is 0.0063091, and the standard deviation is 0.089593,

**Table 3 Detailed parameter values obtained during the implementation of the hybrid SVM-RF model.**

| | |
|---|---|
| Contrast | 0.3059 |
| Correlation | 0.1421 |
| Energy | 0.78623 |
| Homogeneity | 0.93793 |
| Mean | 0.0063091 |
| Standard deviation | 0.089593 |
| Entropy | 3.2051 |
| RMS | 0.089803 |
| Variance | 0.0080177 |
| Smoothness | 0.95913 |
| Kurtosis | 12.2408 |
| Skewness | 1.1048 |
| IDM | 1.2156 |

indicating the variation in pixel values. Entropy, which reflects the randomness in the image texture, is 3.2051. The pixel intensity's root mean square (RMS) is 0.089803, and the variance measures the spread of pixel intensity values, is 0.0080177. The smoothness of the image texture is indicated by a value of 0.95913. The kurtosis, describing the sharpness of the distribution, is 12.2408, while skewness, reflecting the asymmetry of the intensity distribution, is 1.1048. Finally, the inverse difference moment (IDM), which indicates texture uniformity, is 1.2156. These comprehensive parameters help evaluate and optimize the biometric data, contributing significantly to the accuracy and effectiveness of the multi-biometric system for robust identification and authentication.

Figure 6 showcases the performance metrics of the proposed hybrid SVM-RF multi-biometric system, detailing its effectiveness in terms of accuracy, precision, recall, F1-score, and specificity. The system achieved an impressive accuracy of 97.56%, indicating a high level of correctness in identifying individuals across the biometric datasets. The precision rate is 98.89%, reflecting the system's ability to identify true positives among the predicted positive results correctly. The recall, also at 97.37%, demonstrates the system's efficiency in detecting true positives from the actual positive instances, ensuring minimal false negatives. The F1-score, a harmonic mean of precision and recall, is recorded at 98.12%, underscoring the balanced performance of the system in both precision and recall aspects. Lastly, the specificity is 97.92%, highlighting the system's ability to identify true negatives, thereby correctly reducing false positives. These metrics collectively illustrate the robustness and reliability of the hybrid SVM-RF system in multi-biometric identification, showcasing its superior performance compared to conventional methods.

Figure 7 presents a bar graph comparing the performance metrics of the existing and proposed models. The graph demonstrates that the proposed hybrid algorithm
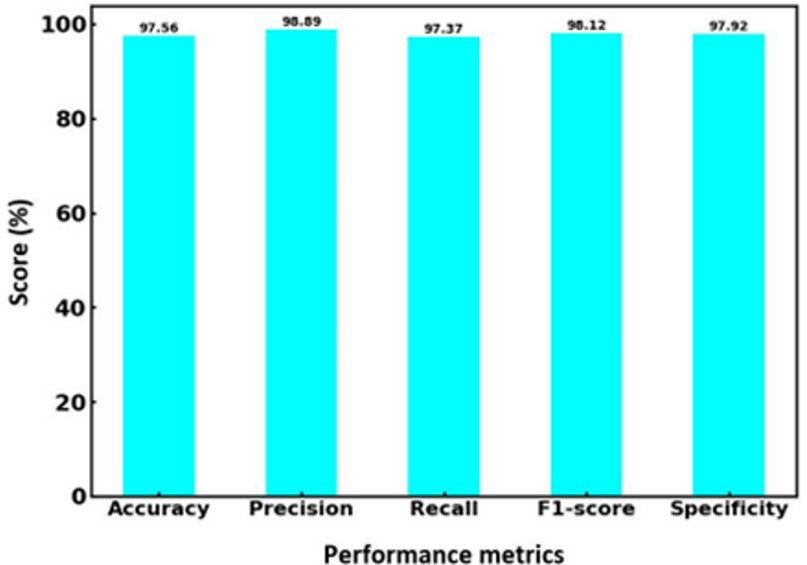

**Figure 6  Performance parameters for hybrid SVM-RF model.**

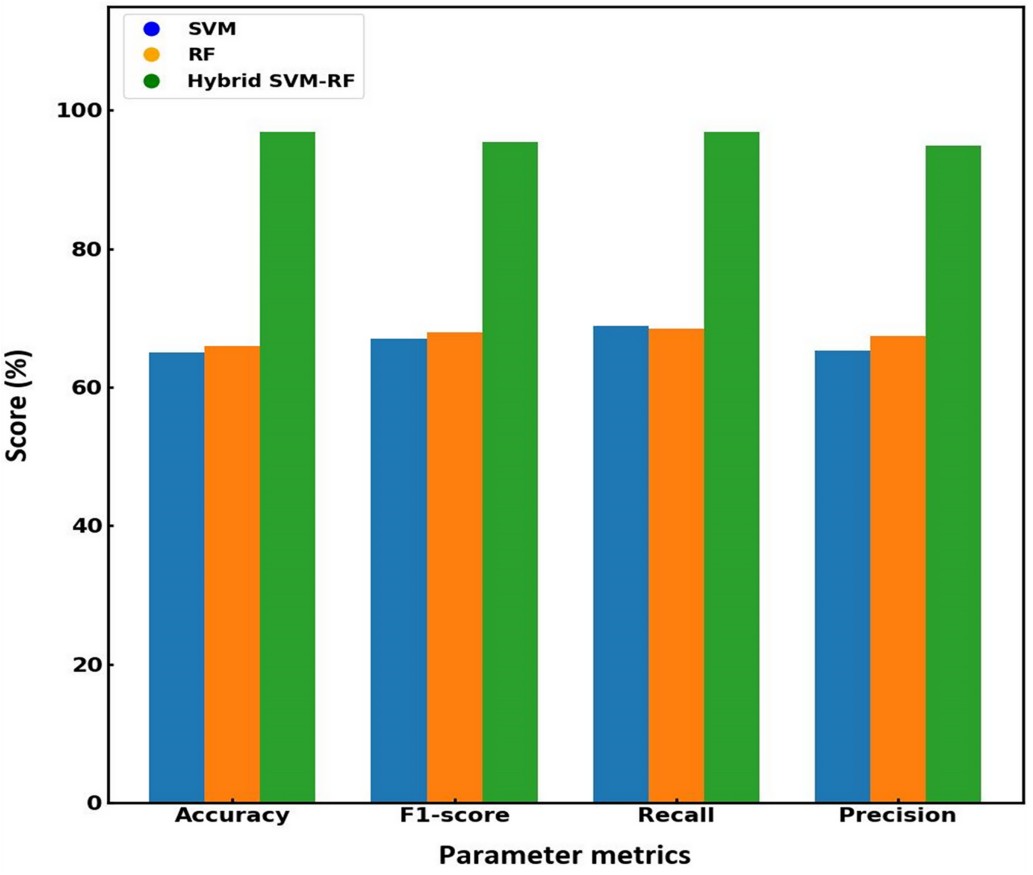

**Figure 7  Graphical illustration of the performance matrix of various models.**

**Table 4 Performance comparison of SVM, RF, and proposed hybrid SVM-RF algorithm.**

| Models | Accuracy (%) | Precision (%) | Recall (%) | F1-score (%) | Loss |
|---|---|---|---|---|---|
| SVM | 54.52 | 65.35 | 68.90 | 65.35 | 0.49 |
| RF | 66 | 67.45 | 68.42 | 67.45 | 0.61 |
| Proposed hybrid SVM-RF | 97.56 | 98.89 | 97.37 | 98.12 | 0.02 |

outperforms SVM and RF across all key metrics. From the visualized data, it is evident that the proposed hybrid SVM-RF model consistently outperforms both the SVM and RF models across all metrics. Specifically, the hybrid algorithm shows an accuracy of 97.56%, superior to that obtained for SVM (54.52%) and RF (66%) models. Regarding loss, the proposed method significantly reduces the error to 0.02, compared to 0.49 for SVM and 0.61 for RF. Precision is highest for the hybrid system at 98.89%, with SVM and RF at 65.35% and 67.45%, respectively. Recall rates for the hybrid method reach 97.37%, while SVM and RF are at 68.90% and 68.42%. Lastly, the F1-score of the proposed system is 98.12%, outperforming SVM (65.35%) and RF (67.45%). This clear dominance of the hybrid SVM-RF model suggests that the integration of the strengths of both SVM and RF into a single framework result in a more powerful and reliable classification tool, making it a preferable choice for applications where high performance across multiple evaluation criteria is critical. Table 4 compares the performance metrics for existing and the proposed models, effectiveness of the hybrid SVM-RF algorithm in enhancing multi-biometric identification accuracy and reliability.

Figure 8 illustrates a comparison of the receiver operating characteristic (ROC) curves and the corresponding area under the curve (AUC) values for the existing SVM and RF and the proposed SVM-RF model. In the comparison, the hybrid SVM-RF model significantly outperforms the SVM and RF models. The AUC for the hybrid model is 0.94, indicating excellent discriminative ability. In contrast, the SVM and RF models achieve lower AUC values of 0.68 and 0.69, respectively, reflecting their comparatively weaker performance. The ROC curve for the hybrid model signifies its superior capability to correctly classify positive instances while minimizing false positives. This enhanced performance highlights the effectiveness of combining SVM and RF in a hybrid approach, which leverages the strengths of both models to achieve higher accuracy and reliability in classification tasks. Therefore, the hybrid SVM-RF model is demonstrated to be the most robust and effective among the models compared, making it a preferable choice in scenarios where precise classification is essential. The comparative analysis of the proposed model with the state-of-the-art approaches is tabulated in Table 5.

While the proposed hybrid multi-biometric system demonstrates significant improvements in accuracy and robustness, it is essential to address computational cost and multimedia data handling to ensure practical applicability. The integration of ML classifiers (SVM and RF) and optimization algorithms (GA and BFO) naturally increases computational demands, particularly during the training phase. To mitigate this,

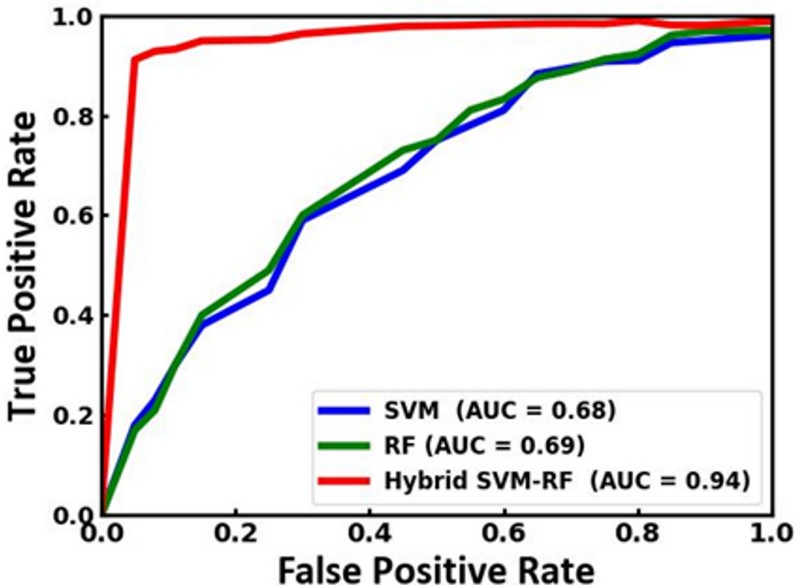

**Figure 8 ROC curve of the SVM, RF and the proposed SVM-RF model.**

**Table 5 Comparison of the proposed SVM-RF model with the existing models towards multibiometric authentication.**

| Model | Modalities | Accuracy | Reference |
|---|---|---|---|
| ResNet-50, VGG16, SVM | Face and iris | 93.33% | *Hasan & Abdulazeez (2024)* |
| Artificial neural network (ANN) | Face, iris and fingerprint | 73.46% | *Lalitha et al. (2024)* |
| Naïve-LLR-GMM | Face & ocular | 93.91 % | *Eskandari & Sharifi (2017)* |
| Modified borda count method | Fingerprint & iris | 85% | *Bala, Gupta & Kumar (2022)* |
| RSA and DNN | Retina, finger and fingervein | 91% | *Srivastava (2020)* |
| HOG and log gabor filter | Iris & Palm-print | 92.23% | *Ramachandran & Sankar (2020)* |
| Mbp-fusion scheme | ECG, fingerprint & face | 92.6% | *Amritha & Aravinth (2020)* |
| Deep reinforcement learning | Multimodal biometrics fusion | 84–93% | *Huang (2022)* |
| Neural and decision level fusion | Iris & fingerprint | 91.5% | *Garg, Vig & Gupta (2016)* |
| Optimal gray wolf optimization (OGWO) | Fingerprint, ear, and palm-print | 91.67% | *Purohit & Ajmera (2021)* |
| Fruit fly optimisation | Fingerprint & iris | 92.23% | *Prakash, Krishnaveni & Dhanalakshmi (2020)* |
| RGB + Entropy network | Face & periocular | 87.41% | *Bala, Gupta & Kumar (2022)* |
| Proposed SVM-RF | Face, iris and fingerprint | 97.56% | This work |

optimization processes are confined to the training stage, while real-time operations leverage pre-trained models to minimize latency. Additionally, parallel processing and dimensionality reduction techniques are employed to reduce runtime computational overhead without compromising performance. The system's modular design supports the integration of diverse biometric modalities, including fingerprint, face, and iris data, and can accommodate multimedia data such as images and video streams. However, the

current implementation focuses primarily on static images. Future enhancements will aim to expand the system's capacity for handling high-resolution and dynamic multimedia datasets. Techniques such as batch processing, adaptive memory management, and hardware accelerations, such as GPU support, will be explored to enhance scalability and processing speed. These considerations ensure the system remains adaptable and scalable, making it a strong candidate for deployment in real-world, high-demand biometric applications. Future work will focus on refining these aspects to fully realize the system's potential for multimedia data handling and computational efficiency.

## CONCLUSION

The hybrid SVM-RF multi-biometric system introduced in this study demonstrates a significant advancement in biometric authentication by effectively integrating face, iris, and fingerprint recognition. The comprehensive methodology employed, which includes feature extraction using Gabor filters, classification through a hybrid SVM-RF approach, and optimization with GA and BFO, has proven to enhance system performance considerably. The proposed system achieves an impressive accuracy of 97.56%, outperforming conventional methods such as SVM and RF, which recorded 54.52% and 66% accuracy, respectively. The experimental results highlight the superior capabilities of the hybrid model in terms of accuracy, precision, recall, and F1-score, underscoring its robustness and reliability in multi-biometric identification tasks. Additionally, the hybrid approach's ability to minimize loss values to 0.02 further underscores its effectiveness compared to the higher loss values observed in traditional algorithms. The successful implementation and optimization of the hybrid system suggest its potential for practical applications, providing a more secure and dependable solution for biometric authentication.

However, the model has some limitations that should be acknowledged. The increased computational complexity from combining multiple classifiers and optimization methods may result in longer processing times, making the system less suitable for real-time applications. The scalability of the system to very large datasets also pose a challenge, as computational demands could grow significantly. Additionally, the system's performance heavily depends on the quality of biometric data, with noisy, low-resolution, or incomplete inputs potentially affecting classification accuracy. While the use of RF helps to mitigate class imbalance, biased results can still occur when certain classes are underrepresented. Moreover, the model does not fully consider variations in environmental conditions, such as changes in lighting, which could impact its reliability in real-world applications. Future research may focus on expanding the number of biometric modalities and further enhancing the security features to address evolving challenges in biometric systems. This study confirms that integrating advanced ML techniques and optimization algorithms can significantly improve the accuracy and reliability of biometric systems, paving the way for more sophisticated and secure identification solutions.

### Funding

The authors received no funding for this work.

### Competing Interests

The authors declare that they have no competing interests.

### Author Contributions

- Sonal conceived and designed the experiments, performed the experiments, analyzed the data, performed the computation work, prepared figures and/or tables, authored or reviewed drafts of the article, and approved the final draft.
- Ajit Singh performed the computation work, prepared figures and/or tables, and approved the final draft.
- Chander Kant performed the experiments, authored or reviewed drafts of the article, and approved the final draft.

### Data Availability

The codes are available in the Supplemental Files.

The fingerprint dataset is available at Kaggle: https://www.kaggle.com/datasets/ruizgara/socofing.

The iris dataset is available at Kaggle: https://www.kaggle.com/datasets/naureenmohammad/mmu-iris-dataset.

The face dataset is available at GitHub: https://github.com/SkyThonk/real-and-fake-face-detection/tree/master/dataset/training.

### Supplemental Information

Supplemental information for this article can be found online at http://dx.doi.org/10.7717/peerj-cs.2699#supplemental-information.

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
