# Peer review of "Optimized hybrid SVM-RF multi-biometric framework for enhanced authentication using fingerprint, iris, and face recognition"

_PeerJ Computer Science, doi:10.7717/peerj-cs.2699_

## Round 0.1 · original submission · Major Revisions

Dear Authors,

Your paper has been reviewed. Based on the reviewers' reports, major revisions are needed before it is considered for publication in PEERJ Computer Science. The issues you have to fix in your revised version of your paper are mainly the following:

1) several claims, such as the following: "The integration of multiple biometric modalities was motivated by the limitations associated with uni-modal biometric systems, such as sensor noise, lack of uniqueness, and high error rates in certain scenarios." must be clearly demonstrated;

2) you have to use appropriate diagrams to clarify the data preprocessing process adopted in your study; 

3) you must compare the performances of your proposed approach with other well-established methods employed to face the same problem, i.e., establish a hybrid multi-biometric system that combines fingerprint, face, and iris recognition;

4) the abstract should be more concise and outline the main objectives.


Reviewer 1 ·

Basic reporting

Overall, the manuscript does follow the format required by PeerJ platform. I do see the updated version where authors have made significant changes based on feedback given by other reviewers especially limitations of the proposed approach do add great value for readers.

Experimental design

Authors must refer to most recent strong & concrete research articles for the topic.
Literature should be categorized for different biometric domains to showcase the need of research of hybrid biometric system.
problem statement is not justified well as it lacks in comparison with anything published before.
Methodology section lacks in technical relevance. Current version is super verbal and vague at some point except the algorithms.

Validity of the findings

Comparative analysis needs more data. authors must compare their proposed approach, the method, results with existing similar research to showcase how their proposed methods/results/approach are better.

Reviewer 2 ·

Basic reporting

This study presents a hybrid multi-biometric system that combines fingerprint, face, and iris recognition. By integrating these three modalities, the system achieves enhanced performance and increased security in real-world scenarios, making it more reliable and robust for practical applications. Two effective machine learning algorithms, Support Vector Machine (SVM) and Random Forest (RF), are employed to achieve this. The model's efficiency is further improved through optimization techniques such as Bacterial Foraging Optimization (BFO) and Genetic Algorithms (GA). The findings reveal that the proposed hybrid model outperforms traditional models, effectively addressing the limitations of conventional identification and verification methods. Furthermore, feature-level fusion and advanced techniques like Gabor filters for feature extraction significantly boost overall performance. The hybrid SVM-RF model achieves a remarkable accuracy of 97.56%, surpassing the individual performances of the SVM (95.5%) and RF (66%) models.

In this version, the authors have revised the details according to the recommendations provided and included the Motivation for the choice of methods used. However, the author should also include references for the claims made, such as the statement: "The integration of multiple biometric modalities was motivated by the limitations associated with uni-modal biometric systems, such as sensor noise, lack of uniqueness, and high error rates in certain scenarios."

Experimental design

The author should present the steps in the data preprocessing process using clear visuals such as images or diagrams. This would help readers better understand this crucial process.

Validity of the findings

The model has certain limitations that need to be considered. Integrating multiple classifiers and optimization techniques increases computational complexity, potentially leading to longer processing times, which may reduce its suitability for real-time applications. Additionally, scaling the system to handle large datasets presents a challenge, as the computational requirements could escalate significantly. How does the system handle real-time constraints, such as processing multiple authentication requests simultaneously? Please ensure that this system can continue to function effectively within a multi-biometric framework for enhanced authentication using fingerprint, iris, and face recognition.

Additional comments

-

Reviewer 3 ·

Basic reporting

The abstract should be more concise and should outline the main objectives
There are some typos and grammatical issues to fix.
The main novelty of the work should be in a separate paragraph in introduction section
Literature work need to enhance by adding more recent works

Experimental design

The proposed meta-analysis does not introduce new insights beyond what is already available in the existing literature, making the contribution appear redundant.
The scope of the study is limited, with insufficient coverage of critical aspects in terms of computation cost, multimedia data handling etc.
The use of machine learning and optimization approaches is not well-defined, and their impact on various algorithms performance is not substantiated with concrete examples or experimental data.
The methodology section lacks clarity regarding the criteria used to assess the relevance and quality of the selected studies, undermining the credibility of the meta-analysis.

Validity of the findings

There is a lack of proper comparison with SOTA methods rather than ML algorithms, which makes it difficult to determine whether the proposed insights provide any significant improvement.
The paper lacks practical validation or simulation results to support its claims about the efficacy
Authors needs to include more samples of the simulated results w.r.t ground truth

---

## Round 0.2 · accepted · Accept

Dear Authors,

Your paper has been re-reviewed. It has been accepted for publication in PEERJ Computer Science. Thank you for your fine contribution.

Reviewer 3 ·

Basic reporting

I am satisfied with the revision

Experimental design

I am satisfied with the revision

Validity of the findings

I am satisfied with the revision